# The Exploration of Chemokines Importance in the Pathogenesis and Development of Endometrial Cancer

**DOI:** 10.3390/molecules27072041

**Published:** 2022-03-22

**Authors:** Jakub Dobroch, Klaudia Bojczuk, Adrian Kołakowski, Marta Baczewska, Paweł Knapp

**Affiliations:** 1Department of Gynecology and Gynecologic Oncology, Medical University of Bialystok, 15-089 Bialystok, Poland; klaudia.bojczuk98@wp.pl (K.B.); adriankolakowski17@gmail.com (A.K.); marta.baczewska@umb.edu.pl (M.B.); knapp@umb.edu.pl (P.K.); 2University Oncology Center, University Clinical Hospital in Bialystok, 15-276 Bialystok, Poland

**Keywords:** endometrial cancer, chemokines, inflammation, cancer treatment, cancer progression

## Abstract

Endometrial cancer (EC) is one of the most frequent female malignancies. Because of a characteristic symptom, vaginal bleeding, EC is often diagnosed in an early stage. Despite that, some EC cases present an atypical course with rapid progression and poor prognosis. There have been multiple studies conducted on molecular profiling of EC in order to improve diagnostics and introduce personalized treatment. Chemokines—a protein family that contributes to inflammatory processes that may promote carcinogenesis—constitute an area of interest. Some chemokines and their receptors present alterations in expression in tumor microenvironment. CXCL12, which binds the receptors CXCR4 and CXCR7, is known for its impact on neoplastic cell proliferation, neovascularization and promotion of epidermal–mesenchymal transition. The CCL2–CCR2 axis additionally plays a pivotal role in EC with mutations in the LKB1 gene and activates tumor-associated macrophages. CCL20 and CCR6 are influenced by the RANK/RANKL pathway and alter the function of lymphocytes and dendritic cells. Another axis, CXCL10–CXCR3, affects the function of NK-cells and, interestingly, presents different roles in various types of tumors. This review article consists of analysis of studies that included the roles of the aforementioned chemokines in EC pathogenesis. Alterations in chemokine expression are described, and possible applications of drugs targeting chemokines are reviewed.

## 1. Introduction

### 1.1. Epidemiology and Classification of Endometrial Cancer

Endometrial cancer (EC) is the most common malignant cancer in women in Europe and the USA and the sixth most common cancer in the women worldwide. The vast majority of EC cases occur in women over 50 years, and the median age of women diagnosed with EC is 63 [1]. The traditional clinical and pathological classification of EC proposed in 1983 by Bokhman distinguished two types of EC: estrogen-dependent type I and estrogen-independent type II. In recent years, this classification has been additionally characterized by molecular phenomena occurring in cancer cells of the uterine mucosa [2]. The current state of knowledge indicates that excessive exposure to estrogens unopposed by the action of gestagens, i.e., nulliparity, obesity, late onset of menopause, and infertility, can be risk factors in EC development [3].

Clinical symptoms of EC course include abnormal uterine bleeding, which is easily discernible at presentation. Because of that typical sign of the disease, approximately 67% of women are diagnosed at the early stage of EC, which has favorable prognosis because of the indolent nature of the cancer. Diagnosis at an advanced stage of EC (stages III and IV, according to the International Federation of Gynecology and Obstetrics (FIGO)), with proclivity for recurrence and worse overall prognosis, occurs less often [4]. Late diagnosis of the advanced disease is the main cause of poor prognosis and worse survival, because advanced EC has a predilection for metastasis formation to the ovaries, lymph nodes, and other areas in the human body [5]. Moreover, in the advanced stage of EC, the efficacy of the anticancer therapy significantly decreases, which includes the problem of chemoresistance. It is vital to investigate the mechanisms that contribute to the development of the advanced EC stage and metastasis formation, which pose the threat of therapy failure. More adequate methods of predicting and treating the advanced stages of this disease should also be found [6].

### 1.2. Implications in EC Classifications and Prognosis

EC is a group of heterogeneous neoplasms. Different types of EC vary in histological features, molecular morphology, and clinical implications. Hence, each classification used to order types of EC entails some generalizations [2]. The EC classifications are based mainly on clinical status, histological subtype, grade, and lymph node invasion, but recently, more attention has been put on diagnosis based on molecular profile. A great number of molecular, inflammatory factors have been extensively studied in order to elucidate which markers are crucial for detection of EC, precursor lesions, or setting the prognosis of early EC [7]. According to recent clinical guidelines, there are some novel and constantly evolving molecular and inflammation markers in endometrial carcinoma that can be used as predictors of dismal prognosis and to determine an adequate personalized treatment approach [2,8]. Since there are no screening tests for EC, an alternative method to detect and assess the type and stage of EC before and after surgery is inevitably needed. The omission of molecular features in EC diagnostics may result in the erroneous risk group assignment and the choice of an inadequate therapeutic strategy [9,10,11]. Moreover, molecular phenomena and inflammation processes that occur in the cancer stroma can influence cancer invasiveness and are crucial for setting adequate diagnosis of EC type.

### 1.3. Molecular Algorithm in EC Diagnosis

According to the latest European Society of Gynecologic Oncology (ESGO) recommendations, the molecular profiling of EC includes four tests: POLE mutation, p53 protein, MSH6, and PMS2. These are necessary to classify the tumor into specific groups: POLE-mutated (POLEmut), p53 abnormal (p53abn), mismatch-repair deficient (MMRd), and nonspecific molecular profile (NSMP). The molecular profile contributes to an assignment to the risk group and setting a prognosis after a diagnosis. For instance, POLEmut ECs are associated with favorable clinical outcome, whereas p53abn tumors have greater tendency towards disease relapse [12]. Molecular analysis is recommended for integration with conventional histopathology in all types of ECs, especially high-grade tumors, in order to optimize a qualification to an adjuvant treatment. However, in low-risk endometrioid cancer, POLE mutation analysis is not essential for diagnostic strategy [13,14]. Moreover, not only those molecules are involved in malignant transformation in endometrial cells. Based on the EC pathogenesis, it is widely acknowledged that inflammation plays a pivotal role in EC development. The prognosis of EC course is also suspected to depend on inflammation factors released in the cancer microenvironment, such as cytokines and chemokines [15]. The numerous data indicate that some chemokine-related reactions can induce resistance to chemotherapy in cancers or be involved in the tumor progression and the metastatic cascade [16,17]. Hence, the detection of those molecules can enhance the accuracy of EC risk assessment and staging and facilitate novel EC classification.

In this review, we provide the current state of knowledge on chemokines’ impact on EC progression and comprehensive understanding of their usefulness in the current classification systems. Finally, we outline that targeting inflammation factors such chemokines in the cancer treatment could have potential clinical application.

### 1.4. Role of Chemokines in Carcinogenesis

It is widely known that inflammation plays a crucial role in malignant transformation [18,19]. The genetic mutations and hormonal alterations observed in EC promote intensification of local inflammatory reactions in the cancer microenvironment. The significant alterations occur in genes that encode proteins crucial for inflammatory processes. As a result of this, the inflammatory cells extensively produce mediators such as cytokines and prostaglandins, the levels of which are augmented in the cancer environment [20]. The majority of inflammation factors are regarded to contribute to carcinogenesis [21]. However, chemokines take precedence over other inflammation factors in the regulation of tumor cells’ activity and adhesion. The numerous data indicate that the cytokine-mediated reactions responsible for immune cells’ migration in tissues are presumed to be a basis of carcinogenesis. For instance, chemokines are suspected to play an essential modulatory role in tumor progression and the metastatic cascade [22,23]. Chemokines can be produced by both cancer cells and leukocytes infiltrating the cancer milieu and can exert effects on immune and nonimmune cells. The hallmark feature of chemokines in tumorigenesis is their ability to regulate lymphocyte migration into the tumor microenvironment, modulate cancer immune response and proliferation, and alter cancer cells’ properties [19]. Recent studies have focused on investigation of chemokines’ influence as a basis of aggressiveness of EC cells.

Chemokines are a family of small, secreted proteins with the ability to induce leukocyte influx to the site of infection. Chemokines exert an influence on cells by interacting with cell surface G protein-coupled heptahelical chemokine receptors [24]. Chemokines play a pivotal role in directing leukocyte migration and immune regulation. In pathological conditions, they also can be secreted and exert an impact on tumor stemlike cells and stromal cells [4,8]. Chemokines are also regarded as a prerequisite for diverse stages of cancer development. Recent studies have proven that there are alterations in chemokines and their receptor expression in some cancers caused by changes in activation of tumor suppressor genes or oncogenes. Precisely, these molecules play a prominent role in tumor growth by inhibiting cell apoptosis [24]. First of all, chemokines are involved in cancer cell aging and inducing cancer cell death. Second, it has been acknowledged that chemokines promote epithelial–mesenchymal transformation (EMT), the most pivotal alteration that initiates metastatic cascade [16]. Moreover, chemokines are presumed to play an important role in cancer-associated angiogenesis and the production of growth factors that trigger neovascularization [25]. Pathways of specific chemokines’ oncogenic influence are summarized in Figure 1.

Studies pertinent to chemokines and their receptors in neoplastic tissues revealed that there is a relevant linkage between the enhanced expression of some proteins in cancer tissues and higher stages of cancer, the presence of metastases in the lymph nodes, and dismal overall survival, e.g., in colorectal cancer [23]. In recent years, novel studies have been conducted that presented the role of chemokines in ovarian, breast, and lung cancer [18,26,27].

## 2. Role of Particular Chemokine-Receptor Axis

### 2.1. CXCL12–CXCR4, CXCR7 Axis

The CXCL12 ligand and its receptors CXCR4 and CXCR7 constitute an axis, which is involved in tumor progression [28]. CXCL12, also known as stromal cell-derived factor 1 (SDF-1), is secreted mostly by the stromal fibroblasts, e.g., in the brain, breasts, liver, lungs, bone marrow, and lymph nodes [29]. CXCL12 may influence cancer cell proliferation, apoptosis inhibition, neovascularization, EMT, and recruitment of tumor-associated macrophages (TAMs) in the tumor microenvironment [22]. The expression of CXCL12 is upregulated and has a remarkable correlation with dismal prognosis in diverse cancers, e.g., breast cancer, pancreatic cancer, ovarian cancer, cervical carcer, and leukemias [30,31,32,33]. In glioblastoma, which has propensity for hazardous invasiveness in patients, the CXCL12–CXCR4 axis contributes to increase cell proliferation and migration, while the suppression of its function decreases cell survival [34]. Much research has indicated an inevitable role of CXCL12 in cancer development by triggering divergent pathways. The CXCL12–CXCR4/CXCR7 signaling pathway exerts an impact on cells by activation of the PI3K/Akt pathway and the MAPK/Erk pathway [35]. CXCL12 induces profound consequences by activation of AKT and ERK pathways, leading to increased NF-κB expression, which results in diminishing apoptotic pathways in cancerous tissues [36]. This hypothesis has been abundantly confirmed by research aimed at inflammation inhibition in cancers. In particular, the study conducted by Jiang et al. highlighted that depletion of CXCR4 suppressed the PI3K/Akt/NF-κβ signaling pathway by triggering apoptosis of human osteosarcoma cells [37]. The CXCL12–CXCR4 axis is also suspected to be involved in initiation of the metastatic cascade [38].

The pronounced expression of CXC12 has a remarkable correlation with the level of vascular endothelial growth factor (VEGF), which is regarded as a vital factor in angiogenesis and cancer invasiveness. Liang et al. reported that CXC12 induced neovascularization by triggering the PI3K/Akt pathway, which resulted in enhanced secretion of VEGF [39].

The expression of CXCL12 and CXCR4 proteins has been proven to be augmented in EC [15,16,40,41]. Liu et al. proved the expression of CXCR4 in 69.23% of EC specimens, which was increased compared with normal endometria [15]. Moreover, other research proved that CXCL12 was expressed in 68% of cases [42]. These data were supported by another recent study conducted on the human endometrial epithelium, which revealed enhanced expression of CXCL12 in 90% of EC specimens [41]. However, Gelmini et al. revealed some discrepancy in CXCL12 expression. Their research revealed a significant decline in CXC12 expression in EC tissue, suggesting that an increase in CXCR4 expression was coupled with the decrease in CXCL12 expression [43].

A vast majority of reports have highlighted the consequences of CXCL12–CXCR4 pathway activation. However, the precise mechanism of triggering CXC12 action in EC is still being widely studied. Teng et al. revealed that cancer-associated fibroblasts release SDF-1α in the EC environment. Their study proved that the hallmark feature of CXCL12 secreted by cancer-associated fibroblasts is its ability to increase cell proliferation, migration, and invasiveness leading to enhanced matrix metalloproteinase (MMP-2 and MMP-9) production in EC cells through the activation of the PI3K/Akt pathway and the MAPK/Erk pathway [40]. The hypothesis that the CXCL12–CXCR4 axis is also involved in tumor growth by enhancing cell proliferation and attenuating cell apoptosis has also been strengthened by a vast number of studies on EC [44,45]. CXCL12 influenced cell proliferation via Akt or ERK1/2 signaling pathway stimulation in endometrial cell lines expressing estrogen receptors (ER) and in cells with expression of PTEN protein (HEC-1A cells) [45]. Additionally, the administration of AMD300, which is a CXC12 antagonist, resulted in a remarkable attenuation of EC cells proliferation and invasion [40]. Moreover, the administration of CXCR4-siRNA and/or CXCR7-siRNA, silencing these receptors, resulted in reductions in EC weight and size in xenografts in nude mice [46]. Therefore, the CXCL12–CXCR4 axis has a crucial influence on the development of EC and may be a potential novel therapeutic target.

The CXCL12–CXCR4 axis is also involved in initiation of the metastatic cascade and cancer invasion. Recent research showed that the administration of anti-CXCR4 antibodies in vivo in nude mice diminished the growth and the number of metastases in the liver, lung, and peritoneum [43]. Moreover, Schmidt et al. highlighted that incubation of EC cell lines with SDF-1 resulted in enhanced cell invasion.

The current research has not proven any correlations between the expression of CXCR4 and histological type [40]. However, there is no unambiguous opinion on the correlation between enhanced expression of CXCL12 and type, stage, and prognosis of EC [41,42]. Concerning the stage of EC, Gelmini et al. reported a marked increase in CXCR4 expression in low-grade compared with high-grade EC, while alterations in CXCL12 expression were not observed when comparing the FIGO stages of EC [43]. On the other hand, Walentowicz-Sadlecka et al. proved that CXCL12 was remarkably increased in advanced-stage EC, but CXCR4 expression was not significantly altered depending on EC staging [41]. In contrast, other research revealed that the expression of CXCR4 and CXCL12 was decreased in the advanced stages of EC [47]. There is a strong need to conduct research to elucidate in which type and stage of EC the expression of CXCL12 and CXCR4 is enhanced, because there have been converse statements about that correlation.

A study conducted on EC tissues by Walentowicz-Sadlecka et al. proved that higher presence of SDF-1 was an independent negative predictor of survival in EC. Moreover, their research highlighted that increased CXCL12 expression was correlated with the profound consequences of increased risk of metastasis and deep myometrial invasion [41]. There was no linkage between CXCR4 expression and metastasis, myometrial invasion, or relapse risk [40,41]. However, another study proved that the abundant presence of CXCL12 in ER-positive EC had a significant correlation with extended progression-free survival compared with EC without ER expression [42]. Hence, the thorough investigation of SDF-1α’s clinical relevance as a prognostic factor in EC is also a compelling area of scientific research.

The role of the CXCR7 receptor, which also binds CXCL12, is still widely discussed [48]. CXCR7, unlike CXCR4, is not a typical G protein-coupled chemokine receptor and does not exert an influence on cells by triggering G protein activation. The impact of CXCR7 on cells is based on heterodimerization with CXCR4 and the augmentation of CXCR4 activity [49].

The expression of CXCR7 is enhanced in several cancers and has a remarkable linkage with poor overall survival in, e.g., breast cancer, cervical carcinoma, and non-small-cell lung cancer cells [50]. The expression of CXCR7 has significant linkage with histological stages and the presence of metastases in pancreatic adenocarcinoma, although the expression of CXCR7 has no prognostic value in this case [51]. Once the chemokine CXCL12 is tethered to the CXCR7 receptor, it activates divergent pathways leading to significant alterations in cancer cell survival, proliferation, and migration. The most pivotal role of CXCR7 in tumor development is associated with stimulation of the AKT signaling pathway and EGFR signaling [52]. Furthermore, pronounced alterations in CXCR7 expression significantly enhance the adhesion ability of prostatic cancer cells by influencing the expression of fibronectin 1, cadherin 11, CD44 antigen, and other molecules involved in extracellular matrix degradation, such as MMP3, MMP10, and MMP14. Moreover, CXCR7 can induce profound consequences on tumor microenvironment angiogenesis by enhancing interleukin-8 and VEGF secretion [53].

The prominent role of CXCR7 in EC progression was suggested by research that revealed that the suppression of CXCR7 action significantly diminished the cell proliferation rate and invasion properties of EC cell lines [48]. Current research has yielded discrepant data on CXCR7 expression in EC. Walentowicz-Sadlecka et al. reported that marked expression of CXCR7 was detected in 100% of EC specimens [41]. However, Gelimini et al. reported that there was no significant alteration in CXCR7 expression in EC compared with normal endometrial tissues [43]. Moreover, the current state of knowledge has not indicated any linkage between CXCR7 expression and the clinicopathological type of EC [48]. CXCR7 expression was not correlated with FIGO stage, the presence of metastases, the depth of myometrial invasion, or the likelihood of EC relapse [41].

Summing up, there is a large body of evidence that the CXCL12–CXCR4/CXCR7 axis contributes to tumor progression and metastatic cascade in EC. However, it has not been established yet which stage of EC is characterized by the highest expression of these molecules. Most research has proven that the expression of CXCL12 and CXCR4 is upregulated in EC in approximately 70% of cases. Therefore, there is a need to assess the expression of these proteins before the administration of the substances targeted at the chemokine network, because their effectiveness depends on the expression levels of CXCL12 and CXCR4, which significantly vary in EC. The clarification of the CXCL12–CXCR4/CXCR7 axis mechanism is indispensable for elucidating its usefulness as an additional pathomorphological test and as a new target of a complementary therapy in EC treatment.

### 2.2. CCL2–CCR2 Axis

CCL2, also known as monocyte chemotactic protein-1 (MCP-1), is a potent chemoattractant for monocytes, macrophages, basophils, T lymphocytes, and natural killer (NK) cells [54]. It is abundantly expressed on the surface of monocytes and facilitates differentiation of monocytes into macrophages. CCL2 binds primarily to the CCR2 receptor, which is present in monocytes and in various tissues in the human body, e.g., blood, brain, heart, kidney, liver, lung, ovary, spleen, and thymus [55]. The hallmark feature of CCL2 is its ability to regulate chemotaxis of monocytes/macrophages, memory T lymphocytes, and NK cells [56]. The vast majority of studies have acknowledged the altered expression of CCL2 in different cancers, e.g., prostatic cancer, colorectal cancer, and breast cancer [57,58,59]. Regarding carcinogenesis, CCL2 is presumed to mediate monocyte influx into tumors and contribute to malignant transformation [59].

The predominant role of CCL2 released by cancer cells and other cells in the cancer environment is based on contribution to metastasis formation. CCL2–CCR2 signaling is of a paramount importance in the intricate process of invasion in the cancer environment and to lymph nodes. The pivotal role of CCL2 is its ability to trigger invasive phenotypes of cancer cells and recruit monocytes to tumor sites [60]. Moreover, CCL2–CCR2 signaling enhances metalloproteinase MMP2 and MMP9 production in human chondrosarcoma and hepatoma cells, which elicits detrimental effects on cancer cell invasion and motility [61,62]. Monocytes can be differentiated into TAMs, which are involved in an increase in cancer growth in diverse cancers. Abundant production of CCL2 caused the growth of prostate cancer, augmented the aggregation of macrophages in vivo, and induced neovascularization in colorectal cancer [23,63]. This recognition was confirmed in a study conducted on mice, which proved that CCL2 acquires the ability to induce breast-tumor metastasis formation in lungs via VEGF [64]. Moreover, augmented CCL2 expression has a remarkable correlation with poor overall survival and is a predictor of cancer recurrence in breast cancer [65]. Interestingly, the inhibition of CCL2 action in metastatic breast tumor significantly diminished the size of metastasis in the lungs and macrophage accumulation in cancer in mice but did not alter the primary breast tumor’s size. On the other hand, this research proved that depletion of CCL2 resulted in the augmented expression of IL-6 and VEGF-A. This recognition suggests that the administration of anti-CCL2 treatment may pose the threat of exerting counterproductive effects on tumor metastasis [66].

Considering the presence of CCL2 in EC, some studies reported that CCL2 expression was significantly enhanced in EC cell lines compared with normal endometrial cells [67]. Considering the grade of the tumor, Pena et al. proved enhanced expression of CCL2 in high-grade compared with low-grade EC [68], whereas Hong-qin et al. did not report any significant alterations in CCL2 expression in EC [48].

The relevance of CCL2 in EC development has not been well assessed in foregoing studies. Previous studies have indicated that CCL2 plays a role in EC progression induced by mutations in suppressor serine/threonine kinase gene (LKB1) [68]. The depletion of the LKB1 gene induces EC invasiveness and leads to the progression to metastatic disease [69]. Pena et al. studied the underlying mechanism that contributes to developing aggressive phenotypes of endometrial adenocarcinomas caused by loss of the LKB1 gene. Their research proved that LKB1 regulates CCL2 production by triggering the AMPK pathway, which leads to increased macrophage migration to the tumor microenvironment. Suppression of CCL2 in EC associated with LKB1 mutations resulted in marked attenuation in cancer development [68].

Moreover, the secretion of CCL2 in cancer milieu is regulated by activating transcription factor 4 (ATF4), which is produced in response to stress conditions in the cancer environment [70,71]. Previous studies have indicated that ATF4 exerts detrimental effects on cancer progression by triggering aggressive phenotypes of cancer cells and inducing treatment resistance. Liu et al. reported that inhibition of ATF4 significantly suppressed EC growth in vivo and led to diminishment in macrophage infiltration. Furthermore, their research proved that CCL2 triggers macrophage recruitment and that ATF4 exerts influence on CCL2 expression and macrophage infiltration in EC. Concluding, ATF4-mediated CCL2 signaling contributes to EC macrophage influx to tumor sites [71]. Therefore, other mechanisms that do not directly affect CCL2 expression, such as ATF4, may also contribute to suppressed EC growth and be a target of therapy.

Interestingly, the significant impact of CCL2 was also detected in other gynecologic malignancies. Penson et al. reported that paclitaxel contributed to a decrease in CCL-2 presence in ovarian cancer patients’ ascites, so it is vital to assess whether CCL2 could be used as a marker of treatment efficacy [72]. Moreover, another in vitro study conducted on human endometrial adenocarcinoma cell lines proved that the production of CCL2 was decreased after incubation with unfractionated heparin [73].

In the summary, the CCL2 signaling pathway plays an inevitable role in EC development, and the overexpression of CCL2 poses the threat of therapeutic failure in gynecologic malignancies. Both the CCL2 chemokine and its regulating factors seem to be attractive targets for EC therapy. On the other hand, numerous clinical studies in other tumors have shown that loss of CCL2 induced increased expression of factors influencing neovascularization. There is a need to start research into therapy targeting CCL2 in EC in order to elucidate the inhibitory effect of this chemokine and investigate whether there are actually side effects of the therapy, such as neovascularization.

### 2.3. CCL20–CCR6 Axis

The CCL20 chemokine is also known as macrophage inflammatory protein (MIP)-3α, Exodus-1, or liver and activation-regulated chemokine (LARC) [74]. At present, only one receptor for this chemokine is known—CCR6 [75]. It is most commonly found in the mucosal sites (lung, intestines), liver, thymus, skin, prostate, and testes [76,77], where it is involved in immunological and structural homeostasis. CCL20 plays a crucial role in inflammation and immunization, mainly through the Th17 lymphocyte pathway with accompanying CCR6 expression [78]. High expression of CCL20 and its receptors was observed in some tumors, demonstrating the role of CCL20 signaling in their development. Some studies have found that CCL20–CCR6 has higher concentrations in cancer cells than in normal tissues and is associated with malignant tumors [79].

The role of the CCL20 chemokine in hepatocellular carcinoma (HCC) is well described; the CCL20–CCR6 axis is considered to be a key factor in tumor progression [80]. It has also been shown that the concentration of CCL20 in this cancer is associated with tumor size, vascular invasion, tumor differentiation, risk of recurrence, and even survival rates of HCC patients [81]. Furthermore, breast cancer patients who showed increased expression of CCL20 had worse survival prognosis [82]. This was caused by the self-renewal of breast cancer stem cells through activation of p65 nuclear factor kappa B (NF-κB) via protein kinase C or p38 mitogen-activated protein kinase [83]. CCL20 expression is also elevated in other cancers, such as pancreatic cancer [84], colorectal cancer [85], and ovarian cancer [86]. Both CCL20 and CCR6 are present in tumor cells [87]. For this reason, self-stimulated cell proliferation may be triggered by the CCL20–CCR6 axis. At the same time, angiogenesis may be activated by the CCR6 receptors present on the endothelium [88], which may then lead to the expression of VEGF in neoplastic cells [89].

One of the most important function of CCL20 in the tumor microenvironment is the infiltration of various cell types. Cells such as dendritic cells (DCs), regulatory T lymphocytes (Treg), and Th17 helper cells are recruited. After recruitment, these cells undergo differentiation under the stimulus of the CCL20–CCR6 axis and influence the tumor microenvironment, leading to an increase or decrease (depending on the type of recruited cell) in the patient’s survival rate [90]. Regulatory T lymphocytes control the autoimmune response and are very common in cancerous tissues. Their role in the neoplastic environment is based on the inhibition of the antitumor autoimmune response [91]. High expression of CCR6 receptors and directional migration to tumor-present CCL20 can also be observed for Treg, as described by Chen et al. [92]. It has also been shown that injection of recombinant murine CCL20 protein into the tumor site promotes tumor progression and increases Treg recruitment, suggesting that the concentration of CCL20 should be considered as a prognostic factor for tumor dissemination [93]. Th17 cells, which are also stimulated by the CCL20–CCR6 axis, exert a similar effect on Treg cells. Increased concentrations of Th17 cells lead to tumor progression through the activation of angiogenesis and immunosuppressive mechanisms [94]. For example, in cervical cancer, a positive correlation between the active phenotype of Th17 cells and CCR6 expression was observed, with a CCR concentration much higher than in healthy cells [95]. The CCL20 chemokine is also responsible for the recruitment of DCs, which enhance the antitumor response of the immune system. CCL20 works by binding to CCR6 receptors present on DCs, which are involved in the recruitment of numerous inflammatory cells and suppression of tumor cells proliferation [96]. However, Bonnotte et al. showed that despite the increase in DC concentration, these inflammatory cells are immunologically immature, and tumor growth is not inhibited [96]. In addition, it seems that the effect of CCL20 on other tumorigenic cells is significant and that CCL20 ultimately supports tumor progression.

The CCL20 chemokine is also crucial in oncological gynecology in neoplasms such as ovarian and EC [97]. The expression of CCL20 has been demonstrated in the HHUA endometrial cell line and more recently on primary endometrial epithelial cells [98,99]. In EC, upregulation of CCL20 has been observed [100,101]. Liu et al. showed that CCL20 expression and secretion was increased in receptor activator for NF-κB (RANK)-overexpressed EC cells treated with RANK ligand (RANKL) in vitro and in vivo. Additionally, CCL20 was shown to accelerate invasion and induce EMT in EC cells [102]. The RANK/RANKL axis induces the secretion and expression of CCL20 in EC cells, which promotes tumor progression and metastasis by EMT. Confirmation of the above results was included in a study describing the influence of high RANK expression on the survival of patients with EC. The results showed that such patients were characterized by reduced survival and more frequent occurrence of metastases [103]. Moreover, Wallace et al. demonstrated the association of CCL20–CCR6 with the inflammatory mediator prostaglandin F-2α (PGF-2α) and its F-prostanoid receptor (FP). Induction of CCL20 by PGF-2α/FP signaling in an endometrial adenocarcinoma cell line was found to be dependent on intracellular signaling by Gq protein, epidermal growth factor receptor (EGFR), extracellular signal-regulated kinase (ERK), calcineurin, and nuclear factor of activated T cell (NFAT). The described mechanism was illustrated by treating FP-rich endometrial adenocarcinoma cells with recombinant CCL20, which resulted in a significant increase in the proliferation of adenocarcinoma cells [104]. In summary, the CCL20 chemokine can be regarded as a potential therapeutic target for reducing the extent of metastasis.

### 2.4. CXCL10–CXCR3 Axis

CXCL10 (interferon (IFN) γ-induced protein; IP-10) is a 10 kDa protein functionally classified as a Th1 chemokine. It binds to the CXCR3 receptor and regulates the immune response by activating inflammatory cells such as T lymphocytes, eosinophils, and monocytes. The activation and guidance of leukocytes to the inflamed area and the persistence of the inflammation can lead to tissue damage [105]. The CXCL10 chemokine can also bind to and activate toll-like receptor 4 (TLR4) [106]. CXCL10 is strongly induced by IFN-γ, IFN-α/β [107], and, to a lesser extent, tumor necrosis factor α (TNFα) [108]. CXCL10 induction requires CXCR3, which is a Gαi protein-coupled receptor. Three isoforms of this receptor are distinguished and bind CXCL9, CXCL10, CXCL11, and CXCL4 [109]. CXCR3 is expressed mainly on activated T lymphocytes (primarily Th1), NK cells, and epithelial cells, which enable migration to the inflammatory site through CXCL10–CXCR3 signaling [110]. In addition to induction of Th1 and NK cells, CXCL10 has been associated with the recruitment of CXCR3(+) CD8(+) T cells to the tumor site. CXCL10 not only recruits these cells but induces the production of granzyme B by them, leading to an enhanced antitumor effect [111]. Barash et al. showed that reduced levels of CXCL10 induced the development of myeloma, while treatment of the CXCL10-Ig fusion protein in mice significantly attenuated the tumor growth. This demonstrates the strong antitumor activity of this chemokine on myeloma cells [112]. Moreover, Barreira da Silva et al. reported that the use of dipeptidyl peptidase 4 inhibitors led to an increase in endogenous CXCL10 concentration and increased the translocation into the tumor of CXCR3-expressing lymphocytes. This led to the suppression and rejection of the experimental melanoma [113]. The CXCL10 chemokine can be used as a prognostic marker of survival, because its increased levels in ovarian cancer, colorectal cancer, and other various cancers were positively correlated with an increased likelihood of survival [114]. However, its action in all neoplasms does not have a clear inhibitory effect. Mulligan et al. showed that in the case of breast cancer, elevated levels of CXCL10 may play a role in tumor invasiveness and progression. The CXCL10–CXCR3 axis may be a point of therapy; however, it relies on its inhibition [115]. On the other hand, Ling et al. showed that enhancement of CXCL10/CXCR3 signaling in liver transplants induced endothelial progenitor cell mobilization and differentiation and new vessel formation, which promote hepatocellular carcinoma relapse and progression [116].

The CXCL10 chemokine may also exert antitumor effects through T-cell, macrophage, or NK-independent angiostatic effects. This effect of CXCL10 was observed in xenograft models of lymphoma, squamous cell carcinoma, and lung adenocarcinoma, in which angiogenesis and tumor growth were significantly reduced [117]. CXCL10 also inhibits angiogenesis associated with basic fibroblast growth factor (bFGF) in advanced uterine EC [118]. Furthermore, activation of IP-10 may lead to inhibition of regrowth or recurrence following intensive treatment of advanced ECs. On the other hand, Degos et al. observed increased levels of CXCL10 in the tumor microenvironment. Increased concentrations of this chemokine lead to the recruitment of NK cells as the main cytotoxic mechanism inhibiting cancer progression. However, NK cell recruitment may not be sufficient because of the alteration in the NK cell profile due to the tumor microenvironment. NK cells are increasingly exhausted as the tumor progresses and lose their antitumor effect [119]. In conclusion, the action of the CXCL10–CXCR3 axis has a different mechanism depending on the type of tumor. In the literature, articles describing mainly antitumor activity are predominant. However, in the case of EC, there is information about the multidirectional action of the CXCL10–CXCR3 axis, which may very effectively interfere with the search for potential targeted therapeutic methods on this axis. 

## 3. Chemokines as a Potential Target of Endometrial Cancer Treatment

The chemokine signaling pathways have sparked an interest in the field of EC therapy in recent years because of their impact on the progression of EC. The vital role of chemokines and their receptors in cancer development mentioned above highlights the promising prospect of antichemokine therapy in EC patients. The current state of knowledge shows that the expression of some chemokines is altered in EC, which elicits a number of effects regarding cancer cell properties. Chemokine signaling can be a potential candidate as immune checkpoint in targeted therapy aimed at inhibiting tumor progression and increasing patient survival. Nevertheless, there are also chemokines that represent autoimmune activity and reduce tumor growth. For this reason, the effects of chemokines on EC should be carefully investigated, as they may vary depending on the type of cancer [120]. Inflammatory cells in the cancer environment are less prone to genetic alterations, so anti-inflammatory approaches have generated increasing interest in chemotherapy-resistant cancer [121]. The administration of chemokine receptor antagonists to conventional therapy seems to mitigate potential problems with resistance to chemotherapy. Recent studies have reported that addition of chemokine inhibitors, for instance, CXCR4 inhibitors, to conventional treatment enhanced the efficacy of therapy in cervical cancer and decreased the likelihood of metastasis presence in animal models [122]. The cornerstone of chemokine-oriented therapy is the wide range of chemokine inhibitors used as adjuvant therapy in combination with conventional chemotherapy in solid tumors treatment. However, their usefulness has been confirmed only in clinical and preclinical trials at present. Only a few of them have been clinically approved in cancer treatment targeting chemokine signaling. Mogamulizumab, which is a monoclonal anti-CCR4 antibody, and AMD3100 (plerixafor), which is a small molecule CXCR4 antagonist, are reserved for hematological malignancies in clinical use [123]. Moreover, recent studies have shown that these chemokine inhibitors can also be effectively used in solid tumors, reducing tumor growth, influencing the phenotype of myeloid cells, and increasing the number of infiltrating NK cells. However, there were serious concerns about the safety of Mogamulizumab because of its ability to deplete Treg cells. In contrast, the development of small molecule CCR4 antagonists with less deleterious side effects, such as AF399/420/1802, has greatly improved the efficacy of cancer vaccines in various preclinical tumor models by preventing the induction of Tregs [124]. The inhibition of the CXCL12–CXCR4/CXCR7 axis has also generated increasing interest in solid tumor therapy, e.g., in glioblastoma [125], ovarian cancer [126], and cervical cancer [122]. However, the relevance of CXCR4 inhibitors in EC treatment was highlighted only in experimental models. Teng et al. demonstrated that AMD3100, a CXCR4 antagonist, markedly decreased the influence of SDF-1α on motility of EC cells, as well as their invasive phenotype and proliferation rate [40]. AMD3100 activity on EC cells was also proved in another in vitro study, which revealed that AMD3100 alleviated CXCL12′s effects on EC cell motility [15]. Additive AMD3100 therapy in EC treatment warrants proper investigation in clinical trials.

Current studies have demonstrated that not only CXCR4 antagonists, but other mediators influence the CXCL12–CXCR4 axis and elicit biological effects such as reduced chemotaxis to cancer sites. Interestingly, kisspeptin-10 markedly suppressed biological effects exerted by CXCL12, leading to attenuation in EC invasiveness [127]. Likewise, preliminary results demonstrated that the use of prodrug of green tea polyphenol (−)-epigallocatechin-3-gallate (Pro-EGCG) in vivo in mice models attenuated the production of CXCL12 in EC, leading to diminished TAM accumulation in EC stroma [128].

The CXCL12–CXCR4 pathway has also been construed as a promising target for treating other gynecologic malignancies. In an animal model, AMD3100 attenuated the progression and the growth of ovarian cancer metastases in the peritoneum compared with the control group [129]. The results of other studies also proved that administration of AMD3100 in mouse model of ovarian cancer significantly diminished metastasis formation in other organs and reduced regulatory T cells’ influx in primary tumors. Moreover, AMD3100 markedly improved the poor clinical outcome in OC [130]. A novel compound therapy that impacts the PD-1–PD-L1 and CXCL12–CXCR4 pathways, inducing immune response, has also seemed to yield beneficial results in ovarian cancer treatment in animal models. Combined therapy consisting of AMD3100 and the anti-PD-1 (aPD-1) antibody resulted in a decline in the growth of ovarian cancer and significantly alleviated ascites progression and increased effector CD8+ T cell aggregation in OC stroma [131]. The antitumor efficacy of low-dose paclitaxel chemotherapy was markedly enhanced in the presence of AMD3100 during incubation in ovarian cancer. This may enable the use of lower doses of taxols in ovarian cancer treatment, leading to satisfactory results while attenuating the adverse effects of taxol therapy. There was a significant reduction in ovarian cancer growth in vitro in human and mouse models under a combined therapy consisting of AMD3100 and taxol compared with models under taxol monotherapy [126]. A therapy regime including radiochemotherapy and plerixafor significantly attenuated cervical cancer progression and limited its dissemination [122]. However, a problem was recognized with the toxicity of using CXCL12–CXCR4 pathway antagonists as an adjunct to conventional therapy. As a result, research began with the development of modified versions of AMD3100, which resulted in the production of AMD-NP-PTX. The new version of this drug is characterized by greater safety and reduced toxicity as well as greater anticancer effect in the treatment of ovarian cancer [132]. It is also worth investigating the action of chemokine antagonists in the context of EC, because reducing the toxicity of such therapy could contribute to approval for the next phase of research.

The inhibition of other chemokine pathways also seems to exert beneficial antitumor effects in EC treatment. Treating EC cells with progesterone and calcitriol reduced CXCL1 and CXCL2 expression, which resulted in limitation of EC dissemination and metastatic process [133]. The unfractionated heparin (UFH) significantly limited the release of CCL2 in EC, which led to alterations in cancer cell properties and therefore to suppression of EC progression [73]. Collectively, these studies have shown that agents that alter chemokine signal transduction pathways have potential antitumor activity in EC treatment.

In conclusion, there have been experimental and preclinical studies in the literature showing the beneficial relevance of EC and other gynecologic malignancies immunotherapy by influencing the chemokine network (Table 1). However, there have been no clinical trials confirming the results obtained in the earlier phases. This may be due to the side effects of monoclonal antibodies observed in clinical trials of drug use in other cancers. Although a synergistic antitumor effect with conventional therapy has been confirmed, insufficient pharmacokinetics and pointless toxicity make it difficult to use this combination in patients. It follows that further research is needed to produce impoverished monoclonal antibodies with lower side effects, which will allow drugs to be investigated in the clinical phase of research.

## 4. Conclusions

EC is frequently detected at the early stage and has a favorable clinical outcome. However, development of advanced EC entails significant clinical implications leading to chemotherapy resistance, dissemination of the disease, and cancer recurrence. Chemokines and their receptors are thought to play a pivotal role in EC progression and metastatic cascade. Current research has demonstrated that the expression of some chemokines is altered in EC, which has a vast number of consequences on tumor biology. Recent studies have proven that chemokines exert diverse effects on EC, which vary depending on the type of chemokine. There is a strong need to assess a precise molecular mechanism and function of each chemokine in the signaling pathways in tumor development and spread in EC. Our review gathers extensive knowledge on the action and interdependence of selected chemokines. The information provided will develop targeted therapies affecting chemokines and related substances and pathways. At the same time, our review informs about doubts related to the modification of chemokine action, which may prevent unexpected side effects. The antitumor efficacy of chemokine antagonists was proved in experimental studies that reported that the addition of chemokine antagonists resulted in decreased tumor growth and attenuated metastasis formation. On the other hand, the suppression of some chemokines induced counterproductive effects by enhancing the expression of factors affecting neovascularization and depletion of Treg cells in some cancers. Moreover, pharmacokinetics and toxicity of chemokine antagonists also hinder the development of combined therapy of chemokine antagonists and chemotherapy. Hence, there is a need to investigate inhibitors affecting the chemokine network that effectively facilitate cancer therapy and have fewer side effects. Therefore, their usefulness in combined therapy with chemotherapeutic drugs in EC should be elucidated in preclinical and clinical trials.

## Figures and Tables

**Figure 1 molecules-27-02041-f001:**
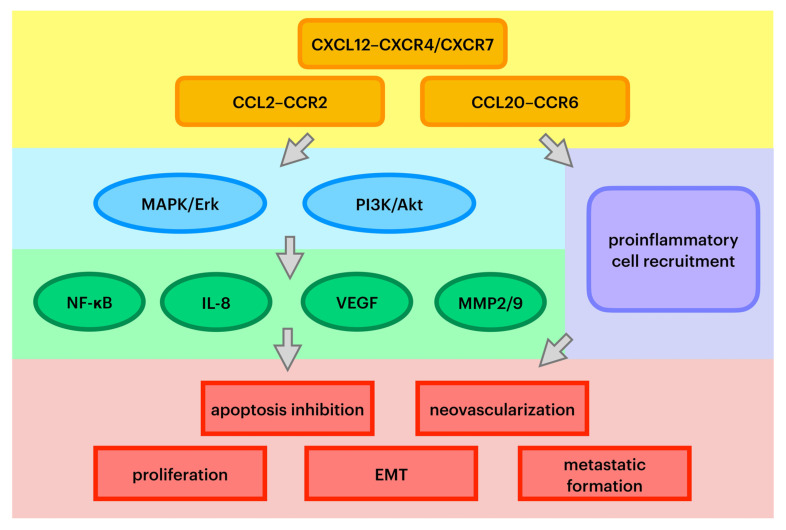
Key pathways of chemokines’ oncogenic effect. MAPK/Erk—mitogen-activated protein kinase/extracellular signal-regulated kinase; PI3K—phosphoinositide 3-kinase; NF-κB—nuclear factor kappa-B; IL-8—interleukin 8; VEGF—vascular endothelial growth factor; MMP2/9—metalloproteinases 2 and 9; EMT—epithelial–mesenchymal transition.

**Table 1 molecules-27-02041-t001:** Summary of chemokine inhibition effects in gynecological malignancies in preclinical studies. * aPD-1—anti-programmed death receptor 1 antibody, ** pro-EGCG—prodrug of green tea polyphenol (−)-epigallocatechin-3-gallate.

Target	Tumor	Inhibitor	Research Phase	Result	References
CXCR4	endometrial cancer	plerixafor	animal model	tumor growth delay, reduction in metastasis formation	[40]
			cell lines	inhibition of cell migration	[15]
	cervical cancer	plerixafor	animal model	tumor growth delay, reduction in metastasis formation	[122]
	ovarian cancer	plerixafor	animal model	overall survival improvement, tumor growth delay, reduction in metastasis formation	[129,130,131]
			cell lines	tumor growth delay	[126]
		AMD-NP-PTX	cell lines, animal model	tumor growth delay, reduction in metastasis formation	[132]
		aPD-1 *	animal model	overall survival improvement, tumor growth delay	[131]
CXCL12	endometrial cancer	kisspeptin-10	cell lines	inhibition of cell migration	[127]
		pro-EGCG **	animal model	angiogenesis inhibition	[128]
CXCL1, CXCL2	ovarian cancer	progesterone, calcitriol	cell lines	reduction in metastasis formation	[133]

## Data Availability

Not applicable.

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
