# Peer review of "The Exploration of Chemokines Importance in the Pathogenesis and Development of Endometrial Cancer"

_molecules, 2022, doi:10.3390/molecules27072041_

Round 1

Reviewer 1 Report

This review is novel in that it focusses on chemokines, in contrast to other recent reviews of molecular characterization of endometrial cancer, which have discussions of all altered molecules.  Furthermore, this article reviews the peer-reviewed literature reporting functional roles of the chemokines and their potential use in endometrial cancer.  The review appears to be up-to-date. 

The main weakness is in the discussion, which appears to be a repetition brief summary of what has already been said, along with some very general and obvious statements, like more research should be done.  The investigators should endeavor to draw some specific conclusions from their review of the literature and provide more in-depth insight about what has been garnered from this review. 

The manuscript could be enhanced by providing a figure or table summarizing the cytokines and their roles in endometrial cancer.  In general, removing some repetition and making it more organized and less a collection of facts would provide improved reader comprehension.

Minor weaknesses are in the grammar, which needs only minor editing in adding commas and "the" and correcting the tense of some verbs. The term "vast" on lines 60 and 102 should be changed to word that means less in comparison to the definition of vast: "unusually great in size or immense".   

In the last paragraph of the Introduction, the term "would like to" should be removed from or changed in lines 94  and 96, because "would like to" means that this is something that the authors may do  in the future, however they have already done it (provide/outline), so substitution words indicating the past or present tense would be more appropriate.

Author Response

Dear Reviewer

Thank you for your time and effort provided to give us a valuable feedback. We really appreciate the suggestions included in you review. Our response to your points are attached below.

Point 1: The main weakness is in the discussion, which appears to be a repetition brief summary of what has already been said, along with some very general and obvious statements, like more research should be done.  The investigators should endeavor to draw some specific conclusions from their review of the literature and provide more in-depth insight about what has been garnered from this review. In general, removing some repetition and making it more organized and less a collection of facts would provide improved reader comprehension.

Response 1: Thank you for pointing this out. We agree with this comment that repetition, brief summary of what has already been said, along with some very general and obvious statements, like more research should be done. We have revised the manuscript to create well- structured draft, corrected repetition and brief summary of what has already been said. We have also come up with some specific conclusions and provided more in-depth insight extensively supported by the scientific papers.

Point 2: The manuscript could be enhanced by providing a figure or table summarizing the cytokines and their roles in endometrial cancer.  

Response 2: As suggested by the reviewer, we have added a figure and table summarizing the cytokines and their roles in endometrial cancer (Figure 1 on page 11 and Table 1 on page 13)

Point 3: Minor weaknesses are in the grammar, which needs only minor editing in adding commas and "the" and correcting the tense of some verbs. The term "vast" on lines 60 and 102 should be changed to word that means less in comparison to the definition of vast: "unusually great in size or immense".

Response 3: Thank you for pointing this out. We agree with the reviewer’s assessment. Accordingly,  we have revised the manuscript, added commas and "the" and corrected the tense of some verbs. We agree with this comment that ‘vast’ is not appropriate and we changes vast to ‘great’ and ‘significant’ on lines 61 and 106 in revised version.

Point 4: In the last paragraph of the Introduction, the term "would like to" should be removed from or changed in lines 94  and 96, because "would like to" means that this is something that the authors may do  in the future, however they have already done it (provide/outline), so substitution words indicating the past or present tense would be more appropriate.

Response 4: Thank you for pointing this out and we agree with this comment. As suggested by the reviewer, we have updated these statements in lines 97 and 99 in revised version and substitute them with present tense. 

Thank you for your tips for our work. We have tried to make all the corrections as stated in the review. In case of any corrections, we are open to further cooperation.

Reviewer 2 Report

This is a well written review article. 

my comments:

  1. The article would benefit from a figure or table depicting major chemokines and pathways in EC highlighting those that are potential therapeutic targets.
  2. a table or box delineating major chemokine axes and their effect on the tumor microenvironment would greately help in the understanding of the molecular impact on carcinogenesis.
  3. regarding the last chapter on therapeutics, it would be critical to report on any potential clinical evidence beyond animal and pre-clinical models, and in case of lacking evidence the authors should elaborate on why relevant research has until now failed to provide a strategy for personalised therapy and why this represents a field with promises for the future.

However, I consider the topic untimely since proteomic research on chemokines has failed, in my opinion to provide exciting perspectives for therapy improvement, as reflected by the general lack of interest for clinical trials, in comparison to more breakthrough research such as genome and transcriptome analysis in the bulk and single-cell levels, especially regarding the Tumor microenvironment.

Author Response

Dear Reviewer, 

Thank you for giving us the opportunity to submit a revised draft of my manuscript titled ‘The exploration of chemokines importance in the pathogenesis and development of endometrial cancer’ to Molecules. We appreciate the time and effort that you and the reviewers have dedicated to providing your valuable feedback on my manuscript. We are grateful to the reviewers for their insightful comments on my paper. We have been able to incorporate changes to reflect most of the suggestions provided by the reviewers. We have highlighted the changes within the manuscript using track changes. All page numbers refer to the revised manuscript file with tracked changes. We have put the responses to your comments below:

Point 1: The article would benefit from a figure or table depicting major chemokines and pathways in EC highlighting those that are potential therapeutic targets.

Response 1:  Thank you for pointing this out. As suggested by the reviewer, we have added or table depicting major chemokines and pathways in EC highlighting those that are potential therapeutic targets (Table 1 on page 13).

Point 2: A table or box delineating major chemokine axes and their effect on the tumor microenvironment would greately help in the understanding of the molecular impact on carcinogenesis.

Response 2: Thank you for pointing this out. As suggested by the reviewer, we have added a figure/box delineating major chemokine axes (Figure 1 on page 11).

Point 3: Regarding the last chapter on therapeutics, it would be critical to report on any potential clinical evidence beyond animal and pre-clinical models, and in case of lacking evidence the authors should elaborate on why relevant research has until now failed to provide a strategy for personalised therapy and why this represents a field with promises for the future.

Response 3: We agree with this comment. As suggested by the reviewer, we have added the additional text at the end of the last chapter in lines 542-548 and 585-593 and 602-610 explaining why relevant research has until now failed to provide a strategy for personalised therapy and why there are no clinical trials in this field.

Point 4: However, I consider the topic untimely since proteomic research on chemokines has failed, in my opinion to provide exciting perspectives for therapy improvement, as reflected by the general lack of interest for clinical trials, in comparison to more breakthrough research such as genome and transcriptome analysis in the bulk and single-cell levels, especially regarding the Tumor microenvironment.

Response 4: Thank you for pointing this out. In general, immunotherapy is currently one of the greatest hopes and branches of medicine especially in cancer therapy and it has surprisingly little use in gynecologic oncology. Since the role of chemokines is well-described and promising in enhancing cell proliferation and attenuating cell apoptosis in vitro studies or in experimental models, this is still an appropriate and promising aim of research. Also there was proven a considerable attenuation of EC invasion and cell proliferation by the suppression of CXCL12/CXCR4 in recent studies. Generally, we think that the general lack of interest for clinical trials is caused by the side effects of drugs targeting chemokine signalling in clinical trials of drug use in other cancers and in earlier phases of trials. Moreover, insufficient pharmacokinetics and toxicity of currently available monoclonal antibodies affecting chemokine network hinder the development of the combined therapy with chemotherapy in patients and do not prove the efficiency of chemokine inhibition in EC therapy what we have added to our review. There is a need to develop bioengineering in this field and look for an adequate methods to inhibit chemokine axes and antagonists and monoclonal antibodies with lower side effects, which will allow drugs to be tested in the clinical phase of research. Furthermore, in our research we provided in part 3 of the review other substances that  indirectly affect the level of chemokines which also contribute to attenuation of EC invasiveness and to the diminished TAMs accumulation in EC stroma. So other substancers are also needed to be investigated in their usefulness in EC therapy. Inflammatory cells in the cancer environment are less prone to genetic alterations so anti-inflammatory approaches generate an increasing interest in chemotherapy-resistant cancer.

Despite our best endeavours to expand the research with the facts that you have mentioned, at the moment this is all available information and research on chemokine targeting therapy trials and the general lack of interest for clinical trials.

Taking all these facts into account there is a need to produce novel therapeutic substances targeting chemokine pathways with lower side effects and investigate other pathways which can indirectly impact the chemokine signalling and be useful in EC therapy.

Thank you for your tips for our work. We have tried to make all the corrections as stated in the review. In case of any corrections, we are open to further cooperation.

Reviewer 3 Report

In the presented review the authors focus their interest and summarize the antineoplastic role obtained through the inhibition of chemokines. In particular they analyze several scientific studies involving CXCL12-CXCL4-CXCR7 axis as well as the role of CCL2-CCL20 or CXCL10-CXCR3 axis and their involvement in the pathogenesis and development of EC.

The review is well write and English is correct.

The introduction section concerning the neoplastic pathophysiology, as well as the role of cytokines is well structured, flowing and the conclusion are extensively supported by the scientific papers and is pertinent to the purpose of this review.

The third section is widely discussed and analyzed; in fact on the basis of the scientific data analyzed, they conclude that the cytokines signaling modulation could be considered as a potential candidate as immune checkpoint in targeted therapy aimed at inhibiting tumor progression and increasing survivor.

So, the aim of their review, ie to demonstrate an increased expression of several cytokines during a neoplastic process and how their inhibition could have an anticancer effect, is spot-on.

Author Response

Dear Reviewer, 

Thank you for giving us the opportunity to submit a revised draft of my manuscript titled ‘The exploration of chemokines importance in the pathogenesis and development of endometrial cancer’ to Molecules. We appreciate the time and effort that you and the reviewers have dedicated to providing your valuable feedback on my manuscript. We are grateful to the reviewers for their insightful comments on my paper. We have been able to incorporate changes to reflect most of the suggestions provided by the reviewers. We have highlighted the changes within the manuscript using track changes. All page numbers refer to the revised manuscript file with tracked changes. We have put the responses to your comments below:

Point 1: In the presented review the authors focus their interest and summarize the antineoplastic role obtained through the inhibition of chemokines. In particular they analyze several scientific studies involving CXCL12-CXCL4-CXCR7 axis as well as the role of CCL2-CCL20 or CXCL10-CXCR3 axis and their involvement in the pathogenesis and development of EC.

Response 1: : Thank you!

Point 2: The review is well write and English is correct.

Response 2: : Thank you!

Point 3: The introduction section concerning the neoplastic pathophysiology, as well as the role of cytokines is well structured, flowing and the conclusion are extensively supported by the scientific papers and is pertinent to the purpose of this review.

Response 3: : Thank you!

Point 4: The third section is widely discussed and analyzed; in fact on the basis of the scientific data analyzed, they conclude that the cytokines signaling modulation could be considered as a potential candidate as immune checkpoint in targeted therapy aimed at inhibiting tumor progression and increasing survivor.

Response 4: : Thank you!

Point 5: So, the aim of their review, ie to demonstrate an increased expression of several cytokines during a neoplastic process and how their inhibition could have an anticancer effect, is spot-on.

Response 5: : Thank you!

Thank you for your tips for our work. We have tried to make all the corrections as stated in the review. In case of any corrections, we are open to further cooperation.

Round 2

Reviewer 2 Report

All my comments have been addressed.